# Resolving Label Uncertainty with Implicit Posterior Models

**Esther Rolf**[*1,6]    **Nikolay Malkin**[*2,6]    **Alexandros Graikos**[3,6]    **Ana Jojic**[4]    **Caleb Robinson**[5]    **Nebojsa Jojic**[6]

[1] University of California, Berkeley, CA, USA
[2] Mila and Université de Montréal, Montreal, QC, Canada
[3] Stony Brook University, Stony Brook, NY, USA
[4] Paul G. Allen School of Computer Science and Engineering, University of Washington, Seattle, WA, USA
[5] Microsoft AI for Good, Redmond, WA, USA
[6] Microsoft Research, Redmond, WA, USA

## Abstract

We propose a method for jointly inferring labels across a collection of data samples, where each sample consists of an observation and a *prior belief* about the label. By implicitly assuming the existence of a generative model for which a differentiable predictor is the posterior, we derive a training objective that allows learning under weak beliefs. This formulation unifies various machine learning settings; the weak beliefs can come in the form of noisy or incomplete labels, likelihoods given by a different prediction mechanism on auxiliary input, or common-sense priors reflecting knowledge about the structure of the problem at hand. We demonstrate the proposed algorithms on diverse problems: classification with negative training examples, learning from rankings, weakly and self-supervised aerial imagery segmentation, co-segmentation of video frames, and coarsely supervised text classification.

## 1 INTRODUCTION

In prediction problems, coarse and imprecise sources of input can provide rich information about labels. Negative labels (what an instance is *not*), rankings (which of two instances is larger), or coarse labels (aggregated by taxonomy or geography) give clues on what the ground truth label of an instance *might* be, but not what it *is* directly. We consider a collection of data samples, indexed by $i$, consisting of observations (features) $x_i$ and corresponding sample-specific *prior beliefs* about their latent label variables, $p_i(\ell)$. This paper proposes algorithms to **resolve the uncertainty in these prior beliefs** by jointly inferring an assignment of target labels $\ell_i$ and a model that predicts $\ell_i$ given $x_i$.

Partial or aggregate annotations and auxiliary data sources are often more widely available and convenient to collect

than "ground-truth" or high-resolution labels, but they are not readily used by discriminative learners. Supervision from probabilistic targets can result in uncertain predictions (§2). Most approaches to resolve these uncertainties involve iterative generation of hard pseudolabels [Zhang et al., 2021] or loss functions promoting low entropy of predictions [Nguyen and Caruana, 2008, Yu and Zhang, 2016, Zou et al., 2020, Yao et al., 2020]. Typically, these approaches are application-specific [Han et al., 2014, Zheng et al., 2021, Bao et al., 2021, Li et al., 2021]. In many settings, fusing weak input data into a probability distribution over classes is a more natural alternative to transforming the weak input into hard labels [Mac Aodha et al., 2019]. Further connections and comparisons to prior work are made throughout this paper and synthesized in §C and §D.

Our key modeling insight (§2.1) is to identify the output distribution of a discriminative model, a feed-forward neural network $q$, with an approximate posterior over latent variables in an *generative* model of features, of which the given prior belief is a part. Bayesian reasoning about the generative model and its posterior makes it possible to learn the inference network *without instantiating the full generative model*, while reaping the benefits of generative modeling: high certainty in the posterior under soft priors and rich opportunities to model structure in the prior beliefs.

Prior beliefs about labels can arise from many sources (§3). We validate the effectiveness of our approach with experiments (§4, §F) on multiple domains and data modalities that highlight: prior beliefs as a natural way to fuse weak inputs, graceful degradation of performance with increasingly noisy or incomplete inputs, and comparison with explicitly generative modeling approaches.

## 2 BACKGROUND AND APPROACH

**Two motivating examples.** Two illustrative examples are shown in Fig. 1. In the first example, the $x_i$ are 784-dimensional vectors representing 28×28 MNIST digits. We

*Accepted for the 38th Conference on Uncertainty in Artificial Intelligence* (UAI 2022).

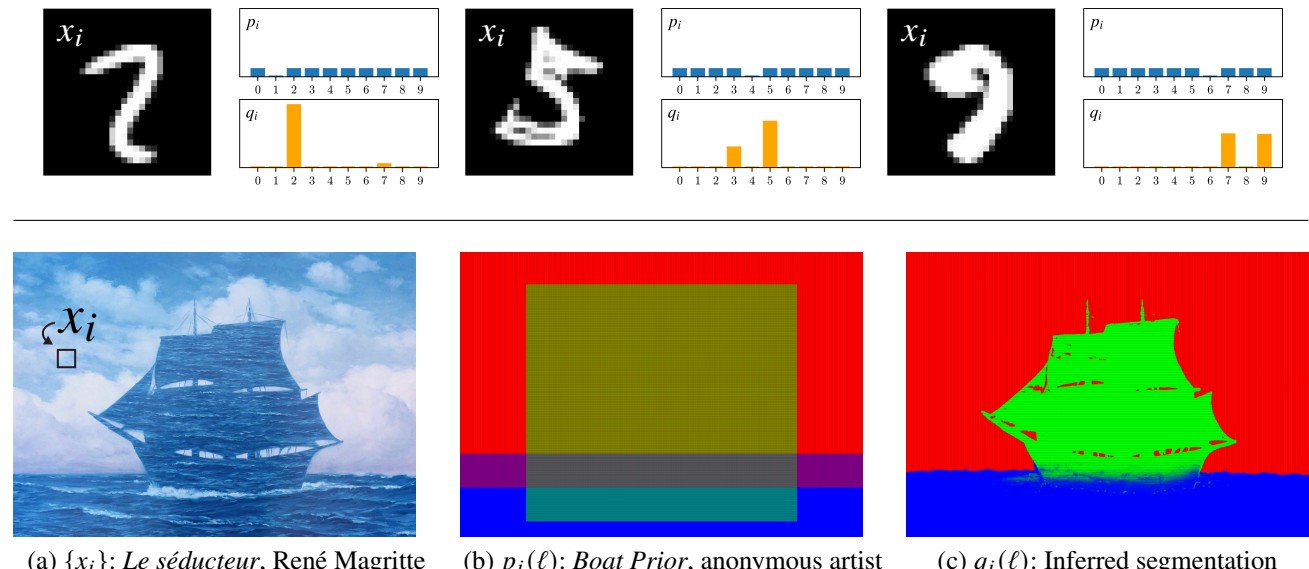

(a) $\{x_i\}$: *Le séducteur*, René Magritte  (b) $p_i(\ell)$: *Boat Prior*, anonymous artist  (c) $q_i(\ell)$: Inferred segmentation

Figure 1: **Above:** Inference of latent MNIST digit classes with negative label supervision using a small CNN trained on the **RQ** criterion (§2.1). **Below:** (a) Joint inference of latent pixel classes in an image. (b) Prior beliefs $p_i(\ell)$ over three classes – sky (red), boat (green), water (blue) – are manually set. (c) A small CNN trained on $(x_i, p_i(\ell))_i$ infers the posterior classes.

aim to infer the digit classes $\ell_i \in \{0, 1, ..., 9\}$ for all images in the given collection based on data in which we are given just one *negative* label per sample, i.e., the prior beliefs $p_i(\ell)$ (top row) are uniform over all classes except for one incorrect class. The procedure described in this paper produces inferred distributions over labels (bottom row) that are usually peaky and place the maximum at the correct digit 97% of the time (see Fig. 3 and §4.1).

In the second example, the observations $\{x_i\}_{i \in \text{pixels}}$ are image patches centered around each pixel coordinate $i$ in a Surrealist painting, with patch size $(11 \times 11)$ equal to the receptive field of a 5-layer convolutional neural network used in our inference procedure. The prior beliefs $p_i(\ell)$ are distributions over 3 classes (sky, boat, water) depending on the coordinate $i$. The joint inference of all labels in this image yields a feasible segmentation despite the high similarity in colors and textures (see §F.4 for more details).

These examples illustrate the problem of training on weak beliefs, which is often encountered in some form in machine learning. Weak supervision, semi-supervised learning, domain transfer, and integration of modalities are all settings where coarse, partial, or inexact sources of data can provide rich information about the state of a prediction instance, though not always a "ground truth" label for each instance. An inference technique that uses weak beliefs as the sole source of supervision needs to estimate statistical links between observations $x_i$ and corresponding latents $\ell_i$. These links should simultaneously be highly confident (i.e., lead to low entropy in the posterior distributions) and explain the varying prior beliefs, which typically have low confidence

(high entropy in the prior distributions).

**Supervised learning on prior beliefs.** Supervised learning models, including many neural nets, are typically trained to minimize the cross-entropy $- \sum_i \sum_\ell p_i^d(\ell) \log q_i(\ell)$ between a "hard" distribution over labels with $p_i^d(\ell) \in \{0, 1\}$ and the distribution $q_i(\ell) = q(\ell|x_i; \theta)$ output by a predictor $q$ using data features $x_i$. This is equivalent to minimizing the KL divergence $\sum_i \text{KL}(p_i^d \| q_i)$, minimized when the two distributions $p_i^d(\ell)$ and $q_i(\ell)$ are equal. Thus, when $p_i^d(\ell)$ is a "softer" prior over latent labels, $p_i(\ell)$, the trained model $q$ will reflect this, and also be highly uncertain.

Transforming soft labels into hard training targets, (e.g. training on $\mathbb{1}[\ell = \arg\max_\ell p_i^d(\ell)]$ ), can introduce the opposite bias. In these cases, the cost would be minimized by predictions with zero entropy, but learning such a prediction function faces difficulty with overconfident labels which are often wrong, and the possibility that certain labels often receive substantial weight in the prior, but never the maximum. These issues are illustrated in Fig. E.3.

**Generative modeling resolves the prior's uncertainty.** The approach to classification problems through *generative* modeling, instead of targeting the conditional probability of latents given the data features, assumes that there is a forward (generative) distribution $p(x_i|\ell)$ and optimizes the log-likelihood of the observed features, $\sum_i \log(x_i) = \sum_i \log \sum_\ell p(x_i|\ell) p_i(\ell)$, with respect to the parameters of that distribution. The posterior under the model $q(\ell|x_i) \propto p(x_i|\ell) p_i(\ell)$ is then used to infer latent labels for individ-

ual data points [Seeger, 2002]. The generative modeling approach does not suffer from uncertainty in the posterior distribution over latents given the input features, even when the priors $p_i(\ell)$ are soft. (Recall that the posterior distributions in a mixture of high-dimensional Gaussians are often peaky even when the priors are flat.)

However, expressive generative models are typically harder and more expensive to train compared to supervised neural networks, as they often require sampling (e.g., sampling of the posterior in variational auto-encoders [VAEs; Kingma and Welling, 2014] and sampling of the generator in GANs [Goodfellow et al., 2014]). Furthermore, the modeling often requires doubling of parameters to express both the forward (generative) model *and* the reverse (posterior) model. And, in case of GANs, the learning algorithms may not even cover all modes in the data, which would prevent joint inference for *all* data points. (See §D for further discussion.)

## 2.1 OPTIMIZING IMPLICIT POSTERIOR MODELS

Suppose that there exists a generative model $p(x|\ell)$ of observed features conditioned on latent labels. Optimization of the log-likelihood of observed features, $\sum_i \log p(x_i) = \sum_i \log(\sum_\ell p(x_i|\ell)p_i(\ell))$, can be achieved by introducing a variational posterior distribution $q(\ell|x_i)$ over the latent variable for each instance $x_i$ and minimizing the free energy (a negated evidence lower bound (ELBO)), defined as

$$-\sum_i \sum_\ell q(\ell|x_i) \log \frac{p(x_i|\ell)p_i(\ell)}{q(\ell|x_i)} \geq -\sum_i \log p(x_i). \quad (1)$$

Minimizing the free energy involves estimating both the forward distributions $p(x_i|\ell)$ and the posteriors $q(\ell|x_i)$.

One could parametrize both $p(x|\ell)$ and $q(\ell|x)$ as functions $p(x|\ell, \theta_p)$ and $q(\ell|x, \theta_q)$ using neural networks, as done by VAEs (although VAEs use continuous latent variables $\ell$ and do not involve sample-specific priors). However, in our algorithms, we only parametrize $q(\ell|x; \theta)$ as a neural network taking input $x$ and producing a distribution over $\ell$. The generative conditional $p(x_i|\ell)$ is defined only on data points $x_i$ and is calculated by minimizing (1) for fixed $q(\ell|x)$, subject to the constraint that $\sum_i p(x_i|\ell) = 1$ for all $\ell$.[1] The optimum is achieved by:

$$p(x_i|\ell) = a_{i,\ell} = \frac{q(\ell|x_i)}{\sum_j q(\ell|x_j)} . \quad (2)$$

Here the generative conditional $p(x|\ell)$ is not fully specified for all values $x$. Rather, it is represented as a matrix of numbers $a_{i,\ell}$ describing the conditional probabilities of different

---

[1]This constraint allows nonzero likelihood under the generative model only for the observed data points $x_i$. The derivation still holds if the assumption is relaxed to $\sum_i p(x_i|\ell) \leq 1$. Subject to this weaker condition, the minimum of free energy is achieved on the boundary of the constraint domain, when $\sum_i p(x_i|\ell) = 1$.

values of $x_i$ given different latent labels $\ell$. The probabilities $p(x_i|\ell)$ are greater for the data points $i$ for which $q(\ell|x_i)$ is more certain, relative to how popular assignment to class $\ell$ is across data points (denominator in (2)).

In our formulation, $q$ plays the role of a variational posterior, but *implicitly*, in a generative model consisting of varying instance-specific priors $p_i(\ell)$ and a complex conditional $p(x|\ell)$ that is never fully estimated, but is instead maximized for the data points studied. The full link between $x$ and $\ell$ is left entirely to the neural network $q$ to capture explicitly.

In variational methods, the free energy (1) is usually rewritten as $\sum_i \text{KL}(q(\ell|x_i)\|r_i(\ell))) - \log p(x_i)$, where $r$ is the posterior of the forward model, i.e., for the points $i$, $r_i(\ell) \propto p_i(\ell)p(x_i|\ell)$. The minimization of free energy then reduces to minimizing the KL divergence between $r$ and $q$.

We define $q_i(\ell) = q(\ell|x_i; \theta)$. After our reduction of $p(x_i|\ell)$ to the auxiliary matrix in (2), the posterior $r$ has the form

$$r_i(\ell) = c_i \cdot p_i(\ell)p(x_i|\ell) = c_i \frac{p_i(\ell)q_i(\ell)}{\sum_j q_j(\ell)} , \quad (3)$$

where $c_i$ are scalars making $\sum_\ell r_i(\ell) = 1$. For each instance $i$ we have two outputs: the direct model outputs of the variational posterior $q_i$ and their *implied posterior* $r_i$, which is computed by multiplying the renormalized model outputs with the provided prior at each instance as in (3). Using these two outputs, we can optimize a single set of model parameters $\theta$ to minimize (1):

$$\min_\theta \sum_i \text{KL}(q_i\|r_i) = \overbrace{\quad}^{\substack{\text{per-}\\\text{instance}\\\text{priors}}} \overbrace{\quad}^{\substack{\text{model output}\\\text{normalized}\\\text{per-class}\\\text{as in Eq. (2)}}} \quad (4)$$

$$\min_\theta \sum_i \text{KL}\left( \underbrace{\left(q(\ell|x_i; \theta)\right)_\ell}_{\substack{\text{model output}\\\text{with input } x_i}} \middle\| \left(c_i \cdot p_i(\ell) \frac{q(\ell|x_i; \theta)}{\sum_j q(\ell|x_j; \theta)}\right)_\ell \right).$$

While (4) optimizes the free energy (1) by minimizing $\text{KL}(q_i\|r_i)$, minimizing $\text{KL}(r_i\|q_i)$ would also find solutions for which the direct model and its implied posterior are close. We propose to optimize either of these two objectives with respect to the model parameters $\theta$ by gradient steps. We iterate over data instances $x_i$ with priors $p_i(\ell)$:

(1) Calculate the distributions $r_i$ in terms of $q_i$ as in (3).

(2) Update the parameters of $q$ with a gradient step:
   • Option **QR**: $\theta \leftarrow \theta - \eta \nabla_\theta \sum_i \text{KL}(q_i\|r_i)$.
   • Option **RQ**: $\theta \leftarrow \theta - \eta \nabla_\theta \sum_i \text{KL}(r_i\|q_i)$.

Gradients of the objectives are propagated to the expression of $r_i$ through $q_i$ (see (4) and Fig. 2). Both losses have a stable point when $q_i = r_i$, and **RQ** reduces to the cross-entropy loss in the case of priors which put all mass on one label (e.g. $p_i(\ell) = \mathbb{1}[\ell = \ell_i]$). A discussion of the relative benefits and limitations of the **QR** and **RQ** losses is given in §B, along with practical considerations for implementation.

```
# log_q : ( batch_size, n_classes ) log-likelihoods from model
# prior : ( batch_size, n_classes ) prior likelihoods

def ce_loss(log_q, prior):
    return -(log_q * prior).sum(1)

def qr_loss(log_q, prior):
    log_r = (log_q.log_softmax(0) + prior.log()).log_softmax(1)
    return (log_q * log_q.exp()).sum(1) - (log_r * log_q.exp()).sum(1)

def rq_loss(log_q, prior):
    log_r = (log_q.log_softmax(0) + prior.log()).log_softmax(1)
    return (log_r * log_r.exp()).sum(1) - (log_q * log_r.exp()).sum(1)
```

Figure 2: Cross-entropy and implicit **QR / RQ** losses in Py-Torch. Here the normalization in (2) is done within batches.

By defining the conditional model $p(x|\ell)$ as an auxiliary matrix of probabilities $a_{i,\ell}$ that is fit to the reverse model $q$ during learning, we avoid parametrizing both directions of the link $\ell - x$ with highly nonlinear models.[2] We thus manage to keep the problem in the realm of training a single feed-forward network $q$ as a predictor of variables $\ell$, but in a way that treats the instance-specific priors $p_i(\ell)$ as they would be in generative modeling.

Next, we discuss the consequences of implicitly modeling the generative model $p$ with an auxiliary distribution. Option **QR** uses the KL distance in the direction it appears in (1) and thus guarantees continual improvements in free energy and convergence to a local minimum (with the exception for the effects of stochasticity in minibatch sampling). Substituting $r_i$ from (3), the free energy (1) becomes:

$$F = \sum_{i,\ell} q_i(\ell) \log \left( \sum_j q_j(\ell) \right) - \sum_{i,\ell} q_i(\ell) \log \left( p_i(\ell) \right) \quad (5)$$

This criterion does not encourage entropy of individual $q_i$ distributions, but of their *average*. The second term alone would be minimized if $q$ could put all the mass on $\arg\max_\ell p_i(\ell)$ for each data point, but the first term promotes diversity in assignment of latents (labels) $\ell$ across the entire dataset. Thus a network $q$ can optimize (5) if it makes different confident predictions for different data points.

To illustrate this, consider the case when all data points have the *same* prior, $p_i(\ell) = p(\ell)$. Then (5) and the **RQ** objective are minimized when $\frac{1}{N} \sum_i q_i(\ell) = p(\ell)$. This can be achieved when $q$ learns a constant distribution $q(\ell|x_i; \theta) = p(\ell)$. But both objectives are also minimized if $q$ predicts only a single label for each data point with high certainty, but it varies in predictions so that the counts of label predictions match the prior.

As demonstrated in Fig. 1 and in our experiments, avoiding

---

[2]Note that the use of an auxiliary matrix $a_{i,\ell}$ is also found in expectation-maximization [EM; Dempster et al., 1977], which also minimizes the free energy. However, in EM, it is the variational posterior $q(\ell|x_i)$ which is optimized as a matrix of numbers $a_{i,\ell}$ only on data points, while the *generative* model $p$ is fully parametrized (see Table D.1).

degenerate solutions is not hard. We attribute this to two factors. First, the situations of interest typically involve uncertain, but varying priors $p_i(\ell)$ which break symmetries that could lead to predictors ignoring the data features $x_i$. Second, the neural networks used to model $q$, and their training algorithms, come with their own constraints and inductive biases. In fact, as discussed in §3 and §F.1, even unsupervised clustering is possible with suitably chosen priors that break symmetry, allowing this approach to be used for self-supervised training. See also §C, §D for more on relationships with other approaches.

In practice, the normalization in (2) is done within batches, rather than across the entire dataset (see Fig. 2). This may be sufficient if batches are large and representative of the diversity in the data. Experiments in §B examine the effect of batch size on performance. While our algorithm is relatively tolerant to moderate batch sizes, performance degrades for small batches, in particular when batches are likely to be missing samples of some classes. Addressing this problem in more general settings is an interesting subject for future work. When intra-batch diversity is an issue, the denominator in (3) may need to be updated in an online fashion or even replaced by a learned parametric estimate.

## 3 SOURCES OF LABEL PRIORS

Having detailed our approach for learning from prior beliefs as weak supervision in §2, we now describe a range of machine learning settings where priors $p_i(\ell)$ emerge. All of these settings are illustrated by experiments in §4 and §F.

**Negative or partial labels (§4.1).** When we are given a set of equally possible labels $L_i$ for each point data point $i$, instead of a single label $\ell_i$, then we set the prior $p_i(\ell) = \frac{1}{|L_i|} \mathbb{1}[\ell \in L_i]$. An extreme example is when one negative label is given and hence can be "ruled out" (Fig. 1).

**Joint labels and learning from rankings (§4.2).** Priors may also come in the form of joint distributions over labels of multiple instances. For example, *ranking supervision* – the knowledge of which example in a pair is greater with respect to an ordering of the labels – gives prior beliefs about *pairs* of labels. Suppose our data is organized into pairs of images of digits $T_j = \{x_{j,1}, x_{j,2}\}$, and for each pair we are told which image represents the digit (0–9) which is greater (or equal). This sets a prior $p(\ell_1, \ell_2)$ over pairs of labels in each pair, represented by either an upper or a lower triangular matrix, depending on which digit in the pair is known to be greater, with all nonzero entries equal to $1/55$.

We assume the underlying generative model has the form $p(x_1, x_2|\ell_1, \ell_2) = p(x_1|\ell_1)p(x_2|\ell_2)$. We aim to fit its posterior model $q(\ell|x; \theta)$. For each pair $T_j$, we have two outputs of the predictor network, $q(\ell_1|x_{j,1})$ and $q(\ell_2|x_{j,2})$, for the two images in the pair. The joint posterior under the genera-

tive model is

$$r_j(\ell_1, \ell_2) \propto p(\ell_1, \ell_2)p(x_{j,1}|\ell_1)p(x_{j,2}|\ell_2) \propto$$
$$\propto \frac{p(\ell_1, \ell_2)q(\ell_1|x_{j,1})q(\ell_2|x_{j,2})}{\sum_j q(\ell_1|x_{j,1})\sum_j q(\ell_2|x_{j,2})}, \quad (6)$$

and we can now use **QR** or **RQ** loss to fit $q(\ell_1|x_{j,1})$ to the marginal $r_j(\ell_1)$ and $q(\ell_2|x_{j,2})$ to $r_j(\ell_2)$.

**Coarse data in weakly supervised segmentation (§4.3, §F.2, §F.4).** We often have side information $z$ associated to each instance $i$ that allows setting the priors $p_i(\ell) = p(\ell|z_i)$ for each point directly by hand. These include situations when we have beliefs about labels for different points, as in the *Seducer* example (Fig. 1). Interesting weak supervision settings also arise in remote sensing (§4.3) and medical pathology (§F.2) applications. For example, in a task of segmenting aerial imagery into land cover classes, we often have coarse labels $c$ associated to large *blocks* of pixels, but not the target labels $\ell$ for individual pixels. If the conditional $p(\ell|c)$ is known, it sets a belief about the high-resolution labels $\ell$ for pixels in a block of class $c$.

**Fusing models and data sources (§4.4, S4.5).** Auxiliary information $z$ may not always come with a known correspondence $p(\ell|z)$. In the land cover mapping problem, auxiliary information includes different modalities and resolutions (road maps, sparse point labels, etc.). While these sources can be fused into a prior by hand-coded rules, the prior may be more accurately set as the output of a model $p(\ell|z_i)$ *trained* on a separate dataset of points $(\ell_i, z_i)$. This is especially useful when the data $x_i$ (imagery) is informative about the latents $\ell_i$ but is prone to domain shift problems, while the auxiliary data $z_i$ does not suffer from domain shift issues but is not sufficient on its own to predict the labels. In a text classification problem, $z_i$ might be the encoding of text $x_i$ by a pretrained language model, and $p(\ell|z_i)$ a noisy distribution over labels given by their likelihoods under the language model as continuations of a prompt.

**Priors for self-supervision (§F.1).** In §2.1 we discussed the pitfalls of using a constant prior $p_i(\ell) = p(\ell)$ for all data points in training models under the **QR** loss as a potential method for unsupervised clustering. However, in §F.1 we give an example of *joint* learning of the posterior model $q$ and an energy model (Markov random field) on the latent labels $\ell_i$ that expresses local structure of labels in an image. This results in unsupervised clusterings that are useful in downstream segmentation tasks. Such an approach is an example of a benefit of generative modeling – the possibility of learning of a parametrized distribution over latents – being inherited by implicit posterior models.

**Priors with latent structure (§F.3).** Implicit posterior modeling allows building hierarchical latent structure into the prior (another benefit of classical generative models),

as we demonstrate in §F.3 on a video segmentation task. The prior is an admixture of possible segmentations with a structure similar to Jojic et al. [2009], but using a set of mask proposals $p(\ell_i|m)$ from a Mask R-CNN model [He et al., 2017], indexed by a latent $m$. The prior is $p_i(\ell) = \sum_m p(\ell_i|m)p(m)$, where $p(m)$, a probabilistic selection of the masks for the admixture in the given frame, is estimated by minimizing the free energy.

# 4 EXPERIMENTS

The experiments in this section and in §F cover a variety of domains, illustrating the sources of label priors listed in §3. The experimental baselines are chosen to reflect the different goals of each experiment. Experiments on classification with negative training examples (§4.1) and learning from rankings (§4.2) serve to illustrate how our algorithm works in different conditions. For experiments on label super-resolution in image segmentation (§4.3, §4.4, §F.1) and text classification (§4.5), self-supervision for image clustering (§F.2), and video segmentation (§F.3), baseline methods provide a comparison by which to benchmark performance, showing that we are reaching or close to state-of-the-art accuracy across these domains with a unified approach.

## 4.1 PARTIAL LABELS IN MNIST AND CIFAR-10

In this experiment, we compare algorithms for learning with partial labels on two 10-class image classification datasets, MNIST and CIFAR-10. To each training example $x_i$, we randomly assign a set $N_i$ of $k$ negative labels, chosen from the 9 labels distinct from the ground truth. The prior $p_i(\ell)$ is set to be uniform over $\ell \notin N_i$ and 0 for $\ell \in N_i$. We vary $k$ from 1 (one negative label per example) to 9 (one-hot prior, full supervision). The data of $k$ negative labels carries $-\log_2(1 - k/10)$ bits of label information; if $k = 1$, $22\times$ less label information than in the fully supervised setting.

For both datasets, the base model $q$ is taken to be a small convolutional network, with four layers of ReLU-activated $3 \times 3$ convolutions with stride 2 and a linear map to the 10 output logits (~33k learnable parameters for MNIST, ~34k for CIFAR-10). We experiment with four training losses:
- **CE:** cross-entropy between predictions $q(\ell|x_i; \theta)$ and the prior $p_i(\ell)$.
- **NLL (union):** negative logarithm of the sum of likelihoods assigned by $q$ to labels in $\ell \notin N_i$, or, equivalently, $\log \sum_\ell p_i(\ell)q(\ell|x_i; \theta)$, as done, e.g., by Jin and Ghahramani [2002], Kim et al. [2019].
- The **QR** and **RQ** losses defined in §2.1.

The **CE**, **NLL (union)**, and **RQ** loss objectives are equivalent when $k = 9$. The **RQ** and **NLL (union)** losses are equivalent when $\sum_i q_i(\ell)$ is uniform over $\ell$ (see derivation in §C), which approximately holds after a sufficient number

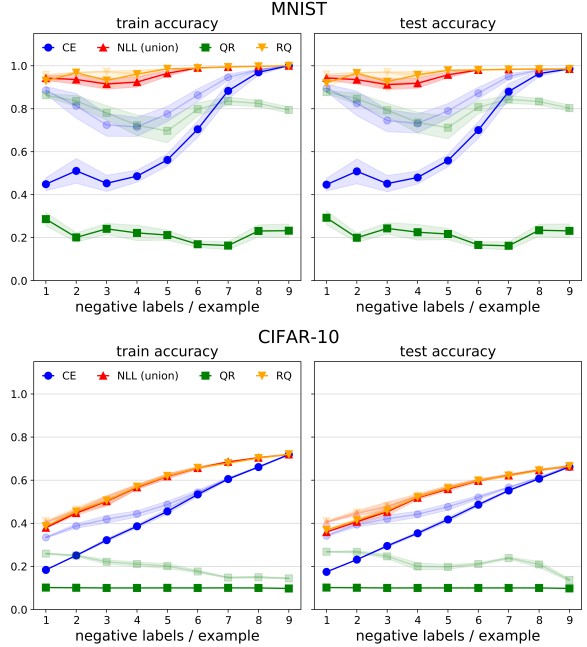

Figure 3: Accuracies of MNIST and CIFAR-10 classifiers trained with varying numbers of negative labels per example; the lighter variant of each color and marker shows the peak accuracy over 300 training epochs. (Average of 10 runs with standard error region.)

of training epochs.

All models are trained for 300 epochs on batches of 256 images with the Adam optimizer [Kingma and Ba, 2014] and a learning rate of $10^{-4}$. After each epoch, we compute the accuracy of the predictor $q$ on the ground truth labels in the train and test sets. Fig. 3 shows the final train and test set accuracies, as well as the maximum accuracies achieved at any epoch. Reported results are averaged over 10 choices of partial label sets and random initializations.

Models trained on **RQ** loss perform best, with the greatest benefit over **CE** seen for very few negative labels. This reinforces the claim in §2 that optimizing the **CE** loss results in uncertain predictions when the priors are highly ambiguous. As expected, the performance of **RQ** and **NLL (union)** is very similar across $k$. We hypothesize that the small advantage of **RQ** over **NLL (union)** loss can be attributed to regularization in early training. Meanwhile, **QR** performs as well as **CE** for very uncertain priors at the peak epoch (light curves), but its predictions degenerate – usually toward uniform predictions – with longer training.

## 4.2 MULTIPLE-INSTANCE SUPERVISION: LEARNING FROM RANKS

We train a CNN of the same architecture as in §4.1 on MNIST, but with the only supervision coming in the form of

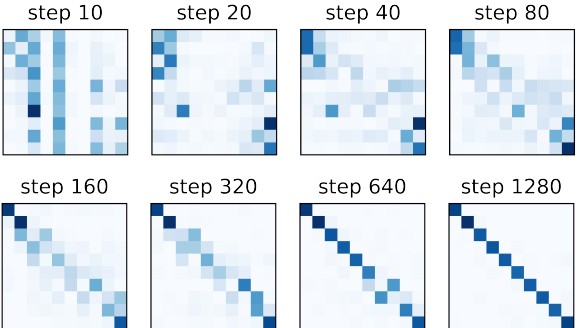

Figure 4: Confusion matrices of MNIST classifiers in the course of training on batches of 128 ranked pairs of digits. The trajectory of convergence to the diagonal shows that uncertainty is first resolved for the digits 0/9, then 1/8, etc.

Table 1: Pixel accuracy and class mean intersection over union on the Chesapeake Land Cover dataset. All models use only coarse NLCD labels as supervision. For our proposed methods, we evaluate both the trained predictor ($q_i$) and the posterior under the generative model ($r_i$). The score of the best overall model is **bolded**.

| Model | PA | | NY | | Chesapeake | |
|---|---|---|---|---|---|---|
| | acc % | IoU % | acc % | IoU % | acc % | IoU % |
| Self-epitomic[a] | **86.2** | 67.6 | 86.4 | 70.5 | 86.3 | 69.7 |
| Hard naïve[b] | 85.3 | 63.0 | 83.6 | 59.8 | 83.6 | 59.7 |
| **QR** ($q$) | 85.9 | 69.3 | 87.3 | 73.0 | 86.4 | 71.1 |
| **QR** ($r$) | **86.2** | **69.9** | **87.9** | **74.4** | **86.8** | **72.1** |
| **RQ** ($q$) | 81.5 | 63.1 | 77.4 | 60.2 | 79.8 | 62.2 |
| **RQ** ($r$) | 81.5 | 63.2 | 77.5 | 60.3 | 79.8 | 62.4 |

[a][Malkin et al., 2020] [b][Malkin et al., 2019]

pairs of images in which it is known which image represents the greater digit. The training set of 60k images is divided into pairs that are fixed throughout the training procedure; each digit appears in exactly one pair. We optimize to match the predictor $q$ with the implicit posterior model (6) using the **RQ** loss. Fig. 4 shows the confusion matrices at initial iterations of training. The learned classifier has 97% accuracy on both training and testing sets, which means that from pairwise comparisons alone, we can group the digit images and place them in order.

## 4.3 LABEL SUPER-RESOLUTION

We benchmark our method's performance on the Chesapeake Land Cover dataset [3], a large 1m-resolution land cover dataset used previously for label super-resolution [Robinson et al., 2019, Malkin et al., 2019]. It consists of several aligned data layers, including: NAIP (4-channel high-resolution aerial imagery at about 1m/px), NLCD (16-class, 30m-resolution coarse land cover labels), and

---

[3]https://lila.science/datasets/chesapeakelandcover

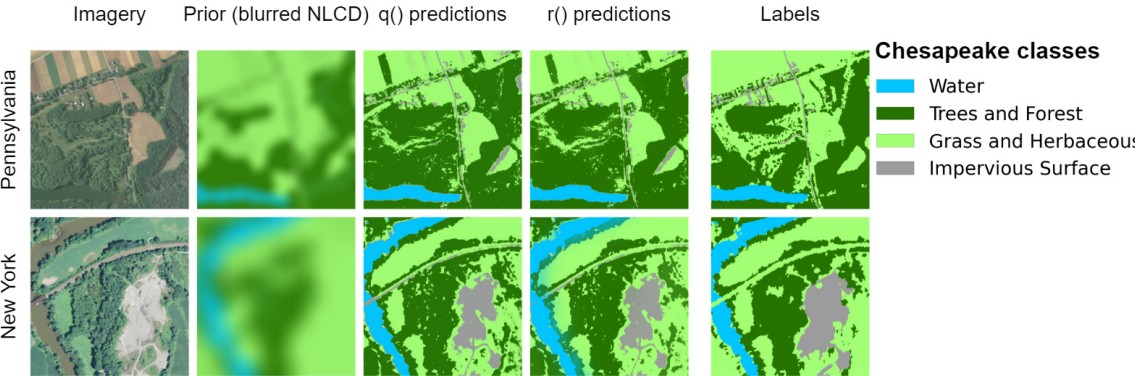

Figure 5: Predictions of models trained with **QR** loss on the NLCD-only prior in the Chesapeake region, shown on regions of 1000×1000 pixels in Pennsylvania and 500×500 pixels in New York.

high-resolution land cover labels (LC) in four classes. The task is to train high-resolution segmentation models, in the four target classes, using only NLCD labels as supervision. The NLCD layer is at 30× lower resolution than the imagery and target labels and follows a different class scheme. Cooccurrence statistics of NLCD classes $c$ and LC labels $\ell$ are assumed to be known (Fig. E.1).

To form a prior over land cover classes $\ell$ at each pixel position, we map the NLCD classes to probabilities over the target LC classes using these known cooccurrence counts and apply a spatial blur to reduce low-resolution block artifacts (Fig. 5, "Prior"). We then train small convolutional networks (receptive field $11 \times 11$) to predict high-resolution land cover from input imagery. We evaluate both the **QR** and **RQ** variants of our approach on the two states that comprise the "Chesapeake North" test set: Pennsylvania (PA) and New York (NY), and the two states combined, after picking hyperparameters based on an independent validation set in Delaware (details in §E.1.3). A depiction of the data and prediction results is given in Fig. 5.

Table 1 compares our algorithms against the algorithmic technique with the best published performance on the Chesapake dataset, self-epitomic LSR [Malkin et al., 2020] and the hard naïve baseline from Malkin et al. [2019]. Self-epitomic LSR, a generative modeling approach that explicitly produces likelihoods $p(x|\ell)$, analyzes small patches of data by making a large number of comparisons between sampled $7 \times 7$ image patches and *all other* image patches. It does not produce a trained feedforward inference model, and the inference procedure is at least an order of magnitude slower than evaluation of our convolutional model. The hard naïve baseline maps the NLCD classes to LC classes based on a given concurrence matrix, then trains a standard semantic segmentation model on these pseudo-labels.

Training on the **QR** loss outperforms (in once case, matches) performance of self-epitomic LSR (Table 1), and the generative model for $p(x|c)$ from (2) is largely consistent with the

epitomic generative model (Fig. E.4). Moreover, our methods handle *batched input*, where self-epitomic LSR trains on one data tile at a time. Similar per-tile approaches have been shown to degrade in performance and exhaust computation capacity when training on multiple tiles [Malkin et al., 2020]). Optimization under an implied generative model has the computational advantage of scaling naturally to large training data while maintaining the benefits of leading generative modeling approaches. (See also §F.2.)

## 4.4 DATA FUSION AND LEARNED PRIORS

In this set of experiments, we augment NLCD with information about the presence of buildings, road networks, and waterbodies/waterways from public sources (see Fig. 6 and §E.1.1). To evaluate the ability of models to generalize to across regions, we use 1m 5-class land cover labels from the geographically diverse EnviroAtlas dataset [Pickard et al., 2015] in four cities in the US: Pittsburgh, PA, Durham, NC, Austin, TX, and Phoenix, AZ. The NLCD-based prior model from §4.3 is augmented with the auxiliary information to obtain a hand-coded prior for each image (see §E.1.2). These types of priors can be made everywhere in the United States, while hard 1m-resolution labels are rarely available.

An alternative to performing local inference under such priors is to simply apply supervised models trained on hard labels elsewhere, hoping that the domain shift is tolerable. Table 2 compares the performance of a model (of the same architecture as in §4.3) trained on Pittsburgh high-resolution data (HR) in each of the three other cities with that of models tuned on the hand-coded prior in each other city. The **QR** method trained on the local handmade prior outperforms the HR model in each evaluation city. This may be attributed to the extra data in each city given to our method in the form of prior beliefs. To isolate this effect, we also compare to a high-resolution model that consumes the prior belief to *input* data, concatenated with the NAIP imagery (HR + aux). While the

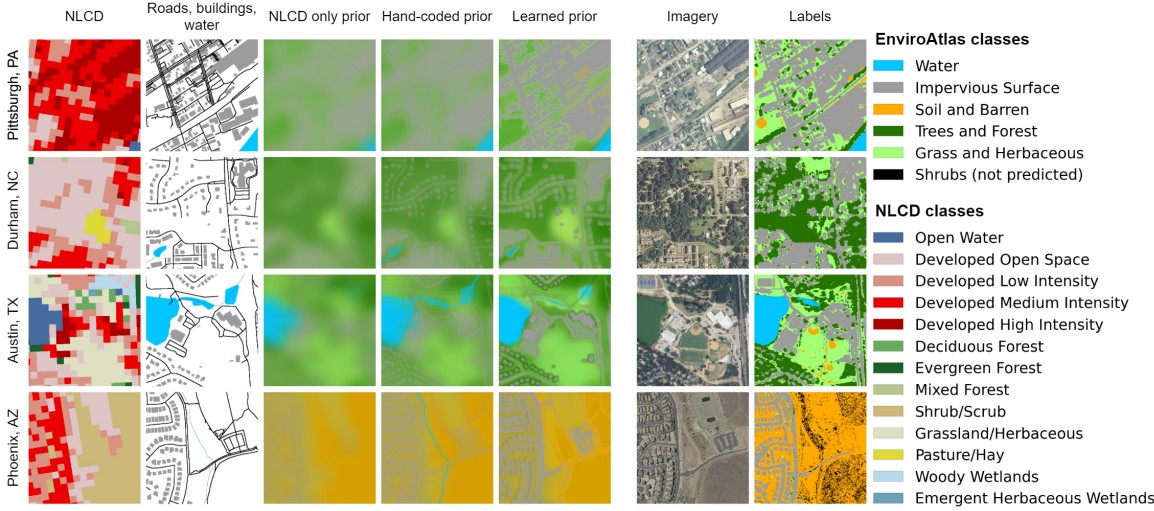

Figure 6: Prior generation for land cover mapping: "NLCD only prior" (§4.3) and "{Hand-coded, Learned} prior" (§4.4).

HR + aux model does increase performance substantially from the HR model with NAIP imagery alone as input, the **QR** model remains the highest-fidelity approach in two of the three cities. These results illustrate that information that generalizes across domains may find its best use within a separate model – to build a prior in our setting – and then used to supervise local inference.

In practice, prior beliefs could be crafted by a domain expert to reflect the uniquities in geographic and structural features for each city. We emulate incorporating such context-specific knowledge by training (on a disjoint set of instances) a neural network that consumes the inputs to the handmade prior function (NLCD and auxiliary map data), and predicts high-resolution labels (Fig. 6, "Learned prior"). Alongside structural interactions between the inputs that generalize across cities (e.g., tree canopy supersedes rivers, roads supersede water), the learned prior captures region-specific knowledge (e.g., buildings in Durham tend to have grass surrounding them and trees farther out, while in Austin, this is reversed, and in Phoenix, riverbeds surrounded by barren land are likely to be dry). Using these tailored prior beliefs during **QR** training tends to increase scores (Table 2).

The final row in Table 2 benchmarks the performance of a high-resolution land cover model trained on imagery and labels over the entire contiguous US [Robinson et al., 2019]. This large model takes NAIP, Landsat 8 satellite imagery, and building footprints as inputs. Small, local models with priors created from only weak supervision outperform the US-wide model in all cities. (See §E.1.4 for details.)

## 4.5 TEXT CLASSIFICATION

This experiment follows the recent work of Mekala et al. [2021] and illustrates the effectiveness of learning on prior

Table 2: Land cover classification experiments for generalizing across cities. In each column, the score of the best model not depending on auxiliary data as input is *italicized* and the score of the best overall model is **bolded**. (A larger set of experimental results is given in Table E.1.)

| Train region | Model | Durham, NC | | Austin, TX | | Phoenix, AZ | |
|---|---|---|---|---|---|---|---|
| | | acc | IoU | acc | IoU | acc | IoU |
| Pittsburgh | HR | 74.2 | 35.9 | 71.9 | 36.8 | 6.7 | 13.4 |
| (supervised) | HR + aux | 78.9 | 47.9 | 77.2 | 50.5 | 62.8 | 24.2 |
| Local | **QR** (q) | 78.9 | 47.7 | 76.6 | 49.1 | *75.8* | *45.4* |
| (hand-coded prior) | **QR** (r) | 79.0 | 48.4 | 76.6 | 49.5 | **76.2** | **46.0** |
| Local | **QR** (q) | *79.0* | 48.7 | ***79.4*** | *51.3* | 73.4 | 42.8 |
| (learned prior) | **QR** (r) | **79.2** | 49.5 | 79.1 | **51.9** | 73.6 | 43.1 |
| Full US[a] | U-Net Large | 77.0 | **49.6** | 76.5 | 51.8 | 24.7 | 23.6 |

[a][Robinson et al., 2019]

beliefs beyond computer vision. We work with a dataset of ~12k New York Times news articles. Each article belongs to one of 20 fine categories (e.g., 'energy companies', 'tennis','golf'), which are grouped into 5 coarse categories (e.g., 'business', 'sports'). The goal is to train text classifiers that predict fine labels, but only the coarse label for each article is available in training.

Some external knowledge about the fine categories is necessary to resolve the coarse labels into fine labels. Past work on this problem [Meng et al., 2018, Mekala and Shang, 2020, Meng et al., 2020, Wang et al., 2021] has trained supervised models on pseudolabels created by mechanisms such as propagation of seed words and querying large pretrained models. On the other hand, Mekala et al. [2021] create training data by sampling additional *features* (articles) from a finetuned version of the large generative language model GPT-2 [Radford et al., 2019] conditioned on fine categories, then tune a classifier based on the almost equally large model BERT [Devlin et al., 2019] in a supervised manner.

Table 3: F1-scores of various models on the coarsely supervised text classification task. The first five rows are taken from Mekala et al. [2021]. The last two rows use the GPT-2 prior defined in §4.5 as weak supervision with cross-entropy and **RQ** loss, respectively (mean of 10 random trials).

|  | Algorithm | Micro-F1 % | Macro-F1 % |
|---|---|---|---|
| pseudolabeling | WeSTClass[a] | 76.23 | 69.82 |
|  | ConWea[b] | 73.96 | 65.03 |
|  | LOTClass[c] | 15.00 | 20.21 |
|  | X-Class[d] | 91.16 | 81.09 |
| pseudodata | C2F[e] | 92.62 | **87.01** |
| GPT-2 prior (trigram features) | prior argmax | 86.33 | 77.61 |
|  | CE | 87.18 | 77.90 |
|  | **RQ** | **93.18** | 84.26 |

[a]Meng et al. [2018] [b]Mekala and Shang [2020] [c]Meng et al. [2020] [d]Wang et al. [2021] [e]Mekala et al. [2021]

We obtain comparable results using an elementary predictor, far less computation, and no finetuning of massive language models (Table 3). We form a prior $p_i(\ell)$ on the fine class $\ell$ of each article $x_i$ by querying GPT-2 for the likelihood of each fine category name $\ell$ compatible with the known coarse label following the prompt "[article text] Topic: " and normalizing over $\ell$. We then divide $p_i(\ell)$ by the mean likelihood of $\ell$ over all articles $x_i$ and renormalize. We represent each article as a vector of alphabetic trigram counts ($26^3$ features, of which only 8k are ever nonzero) and train a logistic regression with the **RQ** objective against this 'GPT-2 prior'. After ten epochs of training (~10s on a Tesla K80 GPU), the trained classifier nears or exceeds the performance of models requiring at least $100\times$ longer to train, even excluding the time to generate any pseudo-training data.

## 5   DISCUSSION AND CONCLUSION

In summary, we found that the generative distribution in a free energy criterion can be left implicit to the minimization process in posterior (discriminative) model training. This allowed us to unite the training of neural networks $q(\ell|x_i; \theta)$ for prediction of labels $\ell$ from features $x$ with the modeling of the prior $p_i(\ell)$, possibly with its own latent structure. Implicit modeling of the conditional generative distributions removes the burden of training accurate (and therefore large or deep) generative models, but still allows natural generative approaches to modeling priors.

Learning a discriminative network $q$ and its implicit posterior model $r$ via the **QR** and **RQ** methods can unify common supervised learning paradigms with realistic label supervision settings, enabling high-fidelity predictions from weak supervision sources carrying far less information. The additional experimental results in §F detail further results for weakly supervised image segmentation, self-supervised learning, and co-segmentation in video data.

Code is available in an accompanying GitHub repository (see §A): `https://github.com/estherrolf/implicit-posterior`.

**Author Contributions**

E.R., N.M., A.G., N.J. jointly conceived the main ideas and their analysis and presentation in this work. E.R. conducted the land cover experiments. N.M. conducted the experiments on negative labels and ranks, text, and lymphocytes and ran the land cover baselines. A.G. conducted the experiments on video tracking and the *Le séducteur* experiments. A.J. and N.J. conducted the experiments on self-supervised image clustering. C.R. helped with compute and storage resources and with implementation of land cover experiments in TorchGeo. All authors collaboratively wrote the paper.

**Acknowledgements**

We thank Anthony Ortiz for helpful feedback during the ideation and writing stages of this work. We also thank the anonymous reviewers for their comments and suggestions.

The main contributions of this work were conceptualized and conducted while E.R. and A.G. were interns at Microsoft Research, Redmond. Computation resources were provided by Microsoft AI for Earth. E.R. additionally acknowledges the support of a Google PhD Fellowship.

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
