# OpenReview forum: "Resolving label uncertainty with implicit posterior models"
_auai.org/UAI/2022/Conference — UAI 2022 Oral_

### Official Review · Reviewer_Y8ti · 2022-04-08

**Q2(1) Originality/Novelty:** 3
**Q2(2) Significance/Impact:** 3
**Q2(3) Correctness/Technical Quality:** 3
**Q2(6) Clarity Of Writing:** 3
**Q6 Overall Score:** 7
**Q8 Confidence In Your Score:** 3

**Q1 Summary And Contributions:**

The paper studies the learning setup where prior beliefs about the labels are available.
A novel training objective is derived that allows learning under weak beliefs.
The effectiveness of the method is demonstrated on a set of diverse problems ( e.g., classification with negative examples, learning from rankings, aerial image segmentation and coarsely supervised text classification).

**Q2 Assessment Of The Paper:**

More detailed information regarding each of these aspects is given below:

**Q2(5) Reproducibility:**

3: Good: Key resources (e.g., proofs, code, data) are available and key details (e.g., proofs, experimental setup) are sufficiently well-described for competent researchers to confidently reproduce the main results.

**Q3 Main Strengths:**

The main strength of the paper is that it bridges the training of discriminative models with the advantages offered by generative models i.e., the ability to leverage on instance-specific priors (that may in turn encode a certain structure). This unlocks a large number of applications where a weak supervision information is available and/or use cases where the weak labels come in various diverse forms (e.g., incomplete labels, partial/sparse labels, coarse vs fine-grained labels etc). This is a big plus.

**Q4 Main Weakness:**

- It seems that the assumption stated in Eq. (2) and the implicit constraint (sum_i p(x_i |l) = 1) need further explanation/motivation. Does it mean that the data samples are kind of "competing" in order to belong to a class? What are the implications for imbalanced classes?

- The denominator in (3) is estimated at the batch level. How much performance cost is incurred because of that? Perhaps an ablation study would shed more light on this question.

**Q5 Detailed Comments To The Authors:**

- In relation to the discussion about "breaking symmetries" at the end of Section 2, have you thought about adding a regularisation term to the training objective to avoid degenerate solutions?

- In Section 4.1 (fourth paragraph). Do you mean Fig. 3? (instead of Fig. 4?)

- Section 4.5 (Text classification). How sensitive is the method to the quality of the prior?

**Q7 Justification For Your Score:**

Please see above.


UPDATE:
I have read the authors' response and I am increasing my score.

**Q9 Complying With Reviewing Instructions:**

1: Yes.

---

### Official Review · Reviewer_s2bo · 2022-04-12

**Q2(1) Originality/Novelty:** 3
**Q2(2) Significance/Impact:** 3
**Q2(3) Correctness/Technical Quality:** 3
**Q2(6) Clarity Of Writing:** 4
**Q6 Overall Score:** 7
**Q8 Confidence In Your Score:** 3

**Q1 Summary And Contributions:**

The presented method suggests a generative model that incorporates observations together with prior beliefs, which can come in various forms (part. labels, rankings, ...). To this end, an implicit posterior model optimization framework is proposed that allows for incorporating a prior belief without leading to overly conservative models. In an exhaustive empirical evaluation comprising various domains, the author could support their claims by showing improved performance over recent baselines.

**Q2 Assessment Of The Paper:**

More detailed information regarding each of these aspects is given below:

**Q2(4) Quality Of Experiments (Optional):**

4: Excellent: The experimental evaluation is comprehensive and the results are compelling.

**Q2(5) Reproducibility:**

4: Excellent: Key resources (e.g., proofs, code, data) are available and key details (e.g., proof sketches, experimental setup) are comprehensively described for competent researchers to confidently and easily reproduce the main results.

**Q3 Main Strengths:**

- Well-written and very intuitively motivated, addresses an important problem
- Elegant prior belief incorporation that is certainly a contribution in its explicitness
- Interesting averaging effect in the result free energy formulation in Eq. (5)
- Very exhaustive empirical evaluation, demonstrating the suitability of the approach in multiple domains / for multiple prior belief “modalities”
- Interesting discussions (also part of the appendix) that relate the approach to previous methods

**Q4 Main Weakness:**

- More thorough related work investigation of previous partial labeling approaches should be considered
- More advanced partial labeling baselines could be considered that reflect the uncertainty in the prior beliefs

**Q5 Detailed Comments To The Authors:**

Major:
- Having just a vague belief about the true outcome is a fundamental problem dimension of partial labeling problems. This is why I would have expected a more rigorous study of previous approaches (e.g., data modeling approaches like Snorkel [1] or loss generalizations such as [2,3], …) and their relations to the proposed method
- Relating to the previous comment, the two considered baselines in Section 4.1 do not capture the uncertainty in the prior adequately. For instance (as also pointed out in the paper in Section 2), CE would assume the prior as exact information rather than an uncertain “guess”. For instance, evidential methods or imprecise probabilistic models could be considered.

Minor:
- The averaging consequence of Eq. (5) reminds me of the (naive) baseline as presented in [1] in the context of partial labeling. Maybe one can relate this to each other
- Title of Section 2.1 is overlengthy

References:
- [1] Ratner, A., et al. VLDB Endowment, 2017. Snorkel: Rapid Training Data Creation with Weak Supervision. (https://arxiv.org/pdf/1711.10160.pdf)
- [2] Hüllermeier, E. IJAR, 2014. Learning from Imprecise and Fuzzy Observations: Data Disambiguation through Generalized Loss Minimization. (https://arxiv.org/pdf/1305.0698.pdf)
- [3] Cabannes, V., et al. ICML 2020. Structured Prediction with Partial Labelling through the Infimum Loss. (https://arxiv.org/pdf/2003.00920.pdf)


**Q7 Justification For Your Score:**

Overall, I appreciate the simplicity yet effectiveness of the proposed method. The paper is well motivated and adequately supported by empirical evidence. Thus, I would like to see it at the conference.

**Q9 Complying With Reviewing Instructions:**

1: Yes.

---

### Official Review · Reviewer_XV9U · 2022-04-12

**Q2(1) Originality/Novelty:** 3
**Q2(2) Significance/Impact:** 3
**Q2(3) Correctness/Technical Quality:** 3
**Q2(6) Clarity Of Writing:** 3
**Q6 Overall Score:** 7
**Q8 Confidence In Your Score:** 3

**Q1 Summary And Contributions:**

The authors discuss an approach to learn from weak supervision by means of a generative model in which the posterior distribution over labels is implicitly defined in terms of a variational distribution which is optimized through a differentiable learning approach. Aside from introducing and motivating the proposed approach, the authors discuss its application in several weak supervision contexts, showing promising results.

**Q2 Assessment Of The Paper:**

More detailed information regarding each of these aspects is given below:

**Q2(4) Quality Of Experiments (Optional):**

2: Fair: The experimental evaluation is weak: important baselines are missing, or the results do not adequately support the main claims.

**Q2(5) Reproducibility:**

3: Good: Key resources (e.g., proofs, code, data) are available and key details (e.g., proofs, experimental setup) are sufficiently well-described for competent researchers to confidently reproduce the main results.

**Q3 Main Strengths:**

- The proposed approach generalizes and unifies several weak supervision problems
- Proposed approach can be implemented using differentiable programming
- Promising results are obtained for different tasks

**Q4 Main Weakness:**

- Experimental analysis could be better motivated: different baselines are used for different experiments and motivation for the considered comparison approach could be better explained

**Q5 Detailed Comments To The Authors:**

The article is interesting as it provides a probabilistic approach to unify several weak supervision problems. The derivation and discussion of the proposed approach seems to be sound and well motivated (though I clarify that I'm not an expert in probabilistic and variational approaches to Machine Learning): the possibility to tackle this problem through differentiable programming is an advantage in computational terms.

 State-of-the-art could be better discussed also in relation with the experimental analysis: how the approach fares (both conceptually and experimentally) in comparison with other approaches that have been proposed in this area and that do not necessarily rely on probabilistic modeling? An example could be the generalized loss minimization frameworks presented in:
Hüllermeier, E. (2014). Learning from imprecise and fuzzy observations: Data disambiguation through generalized loss minimization. International Journal of Approximate Reasoning, 55(7), 1519-1534.
Couso, I., & Dubois, D. (2018). A general framework for maximizing likelihood under incomplete data. International Journal of Approximate Reasoning, 93, 238-260.

Also the selection of baselines, and more in general the discussion of the experimental analysis, could be better motivated: how the baselines were selected e.g. in the partial label learning task on CIFAR-10 and MNIST? Several approaches have been proposed to tackle this problem, thus it would be interesting to understand why the authors selected these two specific methods. Interestingly, in this task the proposed approach does not seem to provide significantly higher accuracy than the approach proposed by Jin and Ghahramani: it would be interesting to understand why this happens and whether the proposed approach offers some particular advantages in this considered task. Nonetheless, the proposed method is more general than the approach proposed by Jin annd Ghahramani, which is obviously a positive point.

**Q7 Justification For Your Score:**

The proposed approach is well motivated and provides a nice generalization and unification of many weakly supervised learning problems considered in the literature. While the experiments and presented results are interesting and promising, discussion of the state-of-the-art is limited and comparison with alternative (also non-probability-based) approaches could be relevant.

**Q9 Complying With Reviewing Instructions:**

1: Yes.

---

### Decision · Program_Chairs · 2022-05-15

**Decision:**

Accept (Oral)

**Comment:**

Meta Review: Reviewers are in consensus that this is a technically strong paper and can have a good impact. The paper provides a unification of several weak supervision problems, and the proposed framework for incorporating prior beliefs allows flexible implementation. The authors are suggested to add more discussions with the recent relevant literature and the additional experiments provided during the rebuttal phase to the final version.